# Physical and Mechanical Properties of Novel Multilayer Bamboo Laminated Composites Derived from Bamboo Veneer

**DOI:** 10.3390/polym14224820

**Published:** 2022-11-09

**Authors:** Xuelian Li, Weizhong Zhang, Wencheng Lei, Yaohui Ji, Zhenhua Zhang, Yifan Yin, Fei Rao

**Affiliations:** 1School of Art and Design, Zhejiang Sci-Tech University, Hangzhou 310018, China; 2Research Institute of Wood Industry, Chinese Academy of Forestry, Xiangshan Road, Haidian District, Beijing 100091, China

**Keywords:** bamboo laminated composite, laminated structure, fiber volume, mechanical properties, water resistance

## Abstract

**Highlights:**

**What are the main findings?**
laminated structure and fiber volume significantly influenced the BLC properties.

**What is the implication of the main finding?**
A reference for selecting an appropriate BLC structure and fiber volume based on ap-plication was provided.The realization of rational allocation of bamboo resources.

**Abstract:**

Compared with wood, bamboo has a special fiber gradient structure. Bamboo fibers have attracted attention as reinforced polymer composites. This study investigated the effects of lamination and fiber volume on the physical and mechanical properties of bamboo laminated composites (BLCs). Six types of BLC were derived by parallel and cross laminating bamboo veneers with high, middle, and low fiber volumes. The results indicated that the laminated structure and fiber volume significantly influenced the BLC properties. Microstructural analysis showed that parallel lamination and low fiber volume were more conducive to resin penetration and enhanced the bonding strength. Both the bending and tensile strengths of the cross lamination were lower than those of the parallel lamination. BLCs made of veneers with high and middle fiber volumes and parallel lamination had the maximum bending and tensile strengths (145.1 and 101.53 MPa, respectively). When tested for water resistance, parallel and cross lamination inhibited expansion in the thickness (*TSR*, 0.56–2.14%) and width (*WSR*, 0.07–1.61%) directions, respectively. Laminated structures and veneers with varying fiber volume contents should be chosen according to the specific application scenarios. This study provides a reference for selecting an appropriate BLC structure and fiber volume based on application.

## 1. Introduction

With increasing environmental awareness, the manufacturing industry is paying increasing attention to sustainable, recyclable, low-cost materials [1,2]. In this scenario, bamboo has great advantages because of its fast growth cycle, sustainability, biodegradability, and cultural attributes. China has the highest abundance of bamboo in the world, with more than 500 species and 6.4 million hectares of pure bamboo forests. Bamboo can be used as fiber feedstock for pulp and paper [3]. In the form of eco-friendly sugarcane bagasse fiber and bamboo fiber, it could replace plastics through a scalable pulp-molding method to develop all-natural biodegradable and low-cost tableware [4]. Bamboo also has great potential as an alternative to wood, as usable bamboo can be harvested in 3–4 years from the time of planting as opposed to timber, which takes decades to mature [5]. However, despite bamboo being an excellent resource for developing a low-carbon economy because of its abundance, wide distribution, rapid growth, high yield, and wide use [6], its sharp taper, thin-walled hollow diameter, uneven structure, and high anisotropy considerably limit its application in engineering structures [7,8,9].

There are various types of composites, such as glass or carbon fiber reinforced polymers and natural fibers [10,11,12]. Bamboo fibers have attracted extensive attention as reinforced polymer composites because of their environmental sustainability, mechanical properties, and recyclability, which are comparable to those of glass fibers [13]. At present, bamboo is widely used to produce bamboo laminated composites (BLCs) such as laminated bamboo lumber (LBL) made from bamboo strips after lengthening with joints [14,15,16], bamboo fiber reinforced epoxy composites derived from bulk natural bamboo [17], and bamboo scrimber composite (BSC), which consists of crushed bamboo fiber bundles saturated in resin and pressed into a dense block [18,19]. The physical and mechanical properties of the composites were significantly improved by impregnation with an appropriate resin and hot pressing. A study on LBL with a density of 0.78 g/cm^3^ exhibited a bending strength or modulus of rupture (*MOR*), modulus of elasticity (*MOE*), and tensile strength (TS) of 126.3 MPa, 11.19 GPa, and 125.9 MPa, respectively [8], demonstrating that LBL is suitable for use in decoration and construction. A bamboo fiber reinforced epoxy composite demonstrated a TS of 168.87 MPa, more than twice that of corresponding values for natural bamboo (68.55 MPa), indicating that it is highly desirable for structural material applications [17]. In addition, a bamboo-fiber reinforced composite (BFRC, a new type of BSC) had a bending strength and *MOR* of up to 253.23 MPa and 26.35 GPa, respectively [20], indicating a remarkable potential for application in outdoor flooring, building materials, and transportation. Advanced processing technology has increased the use of bamboo as a composite material.

Developments in bamboo utilization would allow wood to be replaced with bamboo. Previously, bamboo composites were mainly made of bamboo bundles and strips, with few varieties and low added value. Rotary-cut bamboo veneers from raw bamboo can promote the diversification of bamboo composites, in addition to requiring less labor, producing less dust, having good plasticity, and being more ornamental. They are more widely used for surface decoration than bamboo bundles and strips [21], for example, after dyeing to imitate black walnut [22].

However, there are few studies on BLCs derived from bamboo veneers. Previous research has mainly focused on the preparation of laminated veneer lumber (LVL) and plywood derived from parallel and cross laminated wood veneers, respectively. Several studies have shown that the physical and mechanical properties of LVL and plywood are influenced by the quality of the veneer, wood species, number and order of layers, and the adhesive used for bonding [23,24,25,26]. Moreover, a study on the fiber gradient distribution of bamboo [27] showed that composites made of bamboo veneer were different from those made of wood veneer, with fiber volume content decreasing along the radial direction from the outer to the inner layer [28]. To the best of our knowledge, current research on the influence of fiber volume on the properties of laminated composites is limited. A rational utilization of the special fiber gradient structure of bamboo will achieve better resource allocation than using wood. Therefore, further discussion on the effect of the preparation process on the properties of BLCs is necessary.

In this study, six types of BLC were derived by parallel and cross lamination of phenol formaldehyde (PF) resin-impregnated bamboo veneers with high, middle, and low fiber volumes. The aim of the study was to investigate the influence of laminated structure and fiber volume on the physical and mechanical properties (bending strength and TS) of BLCs and to evaluate the water resistance under rigorous conditions. We explored the possibility of using laminated bamboo in structural and decorative applications, with the goal of expanding the applicability of BLCs for the design of products and structures.

## 2. Materials and Methods

### 2.1. Raw Materials

Bamboo veneers cut to 300 × 300 × 0.50 mm (length × width × thickness) with a moisture content of 7% were purchased from Dongguan Haofeng Decoration Material Co., Ltd, (Dongguan, China). Low-molecular-weight PF resin with 46.0 wt% solid content, 35 cps viscosity, and a pH of 10–11 was supplied by Beijing Dynea Chemical Industry Co., Ltd. (Beijing, China).

### 2.2. Preparation of Bamboo Laminated Composites (BLCs)

Bamboo veneers were made of raw bamboo culms produced by rotary cutting and naturally drying to a moisture content of approximately 7%. They were divided into high, middle, and low fiber volumes (Figure 1). Then, the PF resin was diluted to a 30 wt% solution by adding purified water. Next, the bamboo veneers were immersed in the resin solution for 1 h at 25 °C. Subsequently, they were drip-dried for 7–8 min to remove excess resin and then weighed. The resin-impregnated bamboo veneers were naturally dried to obtain a moisture content of approximately 12 wt%. To form the BLCs, 11 veneers were laminated together either in parallel or cross lamination patterns according to the grain of the veneers. The parallel laminated BLC was defined as type I and the cross laminated form as type II. On this basis, the BLCs made from specific bamboo layers were defined as type I high fiber volume (I-H), middle fiber volume (I-M), and low fiber volume (I-L) and type II high fiber volume (II-H), middle fiber volume (II-M), and low fiber volume (II-L; Figure 1).

Before hot pressing, thickness gauges of 5 mm were placed on both sides of the veneers. The veneers were hot pressed (Carver Inc, Wabash, IN, USA) at a hot-plate temperature of 150 °C for a holding time of 5 min/mm and unloaded at a hot-plate temperature of 30–40 °C. They were then stored indoors at 20 °C and relative humidity of 65%. After two weeks, test samples of varying dimensions were prepared by sanding and cutting the slabs.

### 2.3. Characterization

#### 2.3.1. Microstructure Evaluation

Samples (20 mm × 20 mm × 5 mm) cut from the BLCs were soaked in purified water at 63 °C for 8 h. Then, 20 μm transverse sections of these samples were obtained using a slicer (RM224, LEICA, Wetzlar, Germany), following the methods of Ling [29]. The microstructure of the samples was observed using a digital microscopic system (VHX-6000, KEYENCE, Osaka, Japan) in transmission mode.

#### 2.3.2. Density

Six samples (20 mm × 20 mm × 5 mm) were cut from each type of BLC to test their density according to standard GB/T 17657-2013 [30]. Density was calculated using Equation (1).
(1)$$\rho =\frac{m}{l\times b\times t}\times 1000,\;$$
where m is the quality of the samples, and *l*, *b*, and *t* are the length, width, and thickness of the samples, respectively.

#### 2.3.3. Bending Strength and Modulus

Bending strength and modulus parameters were tested in eight to nine samples (140 mm × 10.50 mm × 5 mm) of each type of BLC. The bending strength and modulus of the BLCs were measured using a universal testing machine (MDW-W50, Jinan Time, Jinan, China). *MOR* and *MOE* were measured parallel to the face grains using the standard GB/T 17657-2013 system in three-point bending mode. The samples were tested at a crosshead speed of 5 mm min^−1^ and a span of 80 mm. *MOR* and *MOE* were calculated using Equations (2) and (3).
(2)$$MOR=\frac{3F_{max}l_{1}}{2bt^{2}},$$
(3)$$MOE=\frac{l_{1}{}^{3}\left(F_{2}-F_{1}\right)}{4bt^{3}\left(a_{2}-a_{1}\right)},$$
where *l*_1_ is the span between two supports; *b* and *t* are the width and thickness of the samples, respectively; *F_max_* is the maximum load on the sample at failure; *F*_2_
*− F*_1_ is the load increase of the straight-line section of the load-deflection curve; and *a*_2_ − *a*_1_ is the deformation increase in the middle of the sample.

#### 2.3.4. Tensile Strength

The tensile strength of the BLCs was measured using a universal testing machine (MDW-W50, Jinan Time, China). Six samples (260 mm × 5 mm × 5 mm) of each type of BLC were used for testing TS according to GB/T 17657-2013, using the equipment for the tensile test. The samples were tested at a crosshead speed of 5 mm min^−1^. TS was calculated using Equation (4).
(4)$$TS=\frac{F_{max}}{bt},$$
where *F_max_* is the maximum load on the sample at failure, and *b* and *t* are the width and thickness of the sample, respectively.

#### 2.3.5. Water Resistance

Six samples (20 mm × 20 mm × 5 mm) of each type of BLC were immersed in water for 8 h in a tank at a constant temperature of 63 ± 1 °C and a pH of 7 ± 1, following the GB/T 17657-2013 standard. Length, width, and thickness were measured every 2 h. The width (*WSR*) and thickness swelling rates (*TSR*) were calculated using Equations (5) and (6).
(5)$$WSR\left(\%\right)=\frac{b_{1}-b_{0}}{b_{0}}\times 100,$$
(6)$$TSR\left(\%\right)=\frac{t_{1}-t_{0}}{t_{0}}\times 100,$$
where *b*_0_ and *t*_0_ are the width and thickness of the pristine samples, respectively; *b*_1_ and *t*_1_ are the width and thickness of the samples after treatment for 2 h, respectively.

## 3. Results and Discussion

### 3.1. Microstructure of the Bamboo Laminated Composites (BLCs)

Figure 2 and Figure 3 show the differences in the microstructure of the BLCs at various magnifications, depicting the bonding interfaces formed by resin and bamboo cells. The bamboo column mainly consisted of vascular bundles surrounded by parenchymatous ground tissue. Parenchyma cells and fiber cells remained substantially intact owing to the relatively low hot-pressing pressure.

The bondline refers to the striped area where the two parts of the bamboo matrix were bonded by resin [31]. In type I-H, the bondline lacked resin, especially the parts containing vascular bundles. Only a small amount of resin was observed in the broken parenchyma, intercellular layers, and cell corners (Figure 2A). The vascular bundles were composed of metaxylem vessels and sheaths of sclerenchyma fibers, which made them dense, and the relatively low hot-pressing pressure was not conducive to resin penetration. The bondline between the parenchyma cells was slightly improved by filling the broken cells and cell corners with resin. Type I-M was comparable to I-H in the amount of resin impregnated into the veneer (Figure 2B). A relatively large amount of resin was observed along the bondline in type I-L because of the few vascular bundles (Figure 2C). These results may be explained by the fact that PF resin permeated, was redistributed, and solidified during the thermo-hydro-mechanical densification of BLCs during the hot-pressing process [32,33,34]. As the penetration of resin in BLCs depends on the diameter of the pit aperture on the bamboo cell wall and the molecular weight distribution of the resin [35,36], the gradient distribution of vascular bundles resulted in varying probabilities of resin filling and/or formation of a covering layer in the cell cavity of the parenchyma cells to form a cell wall polymer. The middle bonding interface of type II-H (Figure 3A), in which most vascular bundles had a severe lack of resin, also indicated that excessive fiber cells negatively affect resin penetration. Moreover, in the cross lamination, the fibers in the profile maintained a long distance along the bonding interface, resulting in a severe lack of resin compared to the other types. For the same reason, there was minimal resin in the middle bonding interface of type II-M (Figure 3B). The inner layer was improved because of the low vascular bundle distribution of II-L (Figure 3C). Therefore, cross lamination magnified the disadvantages of excessive fiber cells that inhibit resin penetration.

### 3.2. Density

The average density of the samples is shown in Figure 4 and ranged from 0.66 (I-L) to 0.77 g/cm^3^ (II-H). For type I, the densities of I-H and I-M were not significantly different but were both higher than that of I-L. Density determines the mechanical properties of bamboo to a great extent and mainly depends on the fiber content, fiber diameter, and cell wall thickness. The density of bamboo increased with the increase in fiber content. Bamboo fiber volume fractions decrease along the radial direction from the outer to the inner layer of the culm wall [27], resulting in bamboo layers of varying densities. This difference was improved by resin impregnation and hot pressing, as the densities of types I-H and I-M demonstrated no significant difference. Among the type II BLCs, the density of type II-H was higher than those of II-M and II-L, with no significant density difference between types II-M and II-L.

### 3.3. Bending Strength and Modulus

The results obtained from the sample bending tests are shown in Figure 5, with a *MOR* ranging from 61.68 (II-M) to 145.1 MPa (I-H) and an *MOE* from 4.33 (II-M) to 10.97 GPa (I-H). Figure 5A,B shows that in both types I and II, the *MOR* of high fiber volume BLC was significantly greater than that of middle and low fiber volume BLC, whereas the *MOR* did not vary significantly between the middle and low fiber volume BLC.

Mechanical properties are generally determined by the fiber volume fraction [37]. The uneven distribution of fiber volume in bamboo makes the strength of bamboo stems vary in distinct layers. A previous study found a significant positive correlation between the fiber volume fraction and bamboo fiber strength [38]. The distribution of the vascular bundle on the outer layer of the bamboo stalk was larger than that on the inner layer (Figure 2 and Figure 3), indicating that the fiber volume content of the former was higher than that of the latter. The higher the fiber volume, the higher the density (Figure 4) and resulting strength of the BLC. Therefore, the bending strength of the outer bamboo layer is significantly higher than that of the inner layer. In addition, the strength of the BLC is not only affected by the fiber volume but is also related to the bonding strength of the layer interface. In types I-M and II-M, the bondline was relatively lacking in resin (Figure 2B and Figure 3B). The bondlines of types I-L and II-L were observed to have relatively substantial amounts of resin (Figure 2C and Figure 3C). The bonding with resin reduces the difference in strength caused by the varying fiber volumes. Therefore, no significant difference was seen between the *MOR* of the middle and low fiber volume BLCs.

Differing laminated structures also influenced the *MOR* and *MOE* of the BLCs (Figure 5A,B). The *MOR* and *MOE* of type I were significantly higher than those of type II in the corresponding parts. As bamboo has no radial transfer structure [39], it grows in height but not in diameter. Thus, bamboo veneers only have longitudinal fiber distribution. When the type I samples, which were composed of parallel veneers along the grain, were subjected to a transverse load during the bending test, all the fibers were subjected to stress, whereas in type II, composed of cross veneers, only half of the fibers were subjected to stress. Therefore, the bending properties of type I were significantly higher than those of type II in the corresponding parts.

The load–displacement curve (Figure 5C) shows that a large displacement was achieved in the samples before failure. Significant variations in the ultimate load and displacement were observed for the specific types of samples. The ultimate load of type I was stronger than that of type II in the corresponding parts. Moreover, compared with type II, the ultimate bearing capacity and deformability of type I was clearly improved. The typical failure modes in the bending test are illustrated in Figure 5D, where cracks were observed in the transverse layers and the samples failed abruptly when one or multiple cracks were developed through the thickness of the transverse layers and propagated to the bondlines. Evident cracks were observed in the bonding interface in type I-H, which indicated poor bonding strength between the veneers and the resin.

### 3.4. Tensile Strength

The tensile test results are shown in Figure 6, with TS ranging from 32.54 (II-H) to 101.53 MPa (I-M). In type I, the TS of I-M and I-H was significantly higher than that of I-L, and there was no significant difference between types I-M and I-H (Figure 6A). The degree of bonding strength with resin varied because of the different fiber volumes (Figure 2). The bamboo culm has a unique, heterogeneous structure that lacks horizontal ray cells, while pits act as the only transverse conduits in the internode [40]. For a lower fiber volume, as per the mechanical interlocking theory, the resin was likely to infiltrate the voids and form spiky structures on the transverse conduits.

Regarding the microstructures of BLCs, compared with type I-H (Figure 2A), more resin penetrated the parenchyma and intercellular layers of I-M (Figure 2B), resulting in a slight difference in TS between types I-H and I-M. For type II (Figure 6B), the TS of II-M was greater than that of types II-H and II-L, but the differences were not significant.

The bondline of II-H (Figure 3A) had almost no resin, indicating a lack of bonding strength with resin. In contrast, a larger amount of resin was observed in the bondline of types II-M and II-L (Figure 3B,C). The bonding with resin reduces the difference in strength caused by the varying fiber volumes.

The various laminated structures significantly affected TS. As shown in Figure 6A,B, the TS of type II sharply decreased compared with that of type I. In the tensile test of type I, the TS paralleled the direction of fiber distribution. The cross lamination in type II resulted in a loss of almost half of its strength. Thus, the TS of type I was significantly higher than that of type II in the corresponding parts.

After the critical load was reached, the load of the samples decreased in a phased manner (Figure 6C), indicating that the bonding interface of the BLC failed in each layer under the tensile load (Figure 6D). Distinct sample types exhibited significant variations in ultimate load and displacement. The ultimate load of type I was significantly higher than that of type II in the corresponding parts. The results indicated that, compared with cross lamination, parallel lamination further strengthened the TS of the samples. Figure 6D shows that the failure modes of types I and II under tensile stress were parallel to the grain. The tensile failure of type I-H resulted from the debonding of the composed laminae. In contrast, the fracture of the other samples was mainly due to crack propagation traveling from the point of fiber breakage to the bonding interface.

### 3.5. Water Resistance

Figure 7 shows the *TSR* and *WSR* of the BLCs after an 8 h water treatment. The final *TSR* ranged from 0.56 (I-M) to 2.14% (II-H), and the final *WSR* ranged from 0.07 (II-L) to 1.61% (I-H). For type I, a slight difference was found between the *TSR* of the three samples. However, for type II, the *TSR* of II-H was significantly higher than that of types II-M and II-L, but the difference between II-M and II-L was not significant. This might be caused by the conspicuous lack of resin at the bondline of II-H (Figure 3A), as the weak bond strength resulted in decreased water resistance.

The distinct laminated structure of the BLC significantly affected its water resistance. The *TSR* of type II was significantly higher than that of type I. Compared with type II, resin distribution in the bond line of type I was more uniform (Figure 2 and Figure 3). Type I was less permeable to water, possibly because of the relatively compact internal structures, the smaller and more punctured pores, and the linear cracks blocked by the resin. In contrast, in type II, the fibers in the profile were well separated along the bonding interface because of cross lamination (Figure 3), resulting in a decrease in bonding strength and water resistance. Therefore, the *TSR* of type II was significantly higher than that of type I.

Types I-H and I-L had the maximum and minimum values of *WSR*, respectively. With an increase in the fiber volume content, the bonding strength becomes weaker, resulting in a decrease in water resistance. Types II-M and II-L had the maximum and minimum values for type II, respectively. After 4 h, the *WSR* of II-H and II-L showed a downward trend. This might be because cross lamination can reduce the deformation caused by anisotropy in the width direction. The difference of *WSR* was reduced by cross lamination which inhibited contraction and deformation in the width. In addition, by increasing temperature and time of exposure, the phenomenon of absorption and expansion of bamboo can be reduced [41].

The *WSR* of type I was significantly higher than that of type II. As in wood, bamboo has the anisotropy of dry shrinkage and wet expansion, with dryness shrinkage being greatest in the chord and radial directions, respectively. Dryness shrinkage in the axial direction is negligible. In the cross lamination of type II, the adjacent veneer force direction was vertical, and adjacent forces canceled one another, reducing contraction expansion and deformation. In practical applications, cross lamination can reduce the deformation caused by the anisotropy of the BLCs.

## 4. Conclusions

The aim of this study was to investigate the physical and mechanical properties of BLCs prepared using parallel (type I) and cross lamination (type II) of bamboo veneers with high, middle, and low fiber volumes. The physical properties of the BLC samples were significantly affected by the laminated structure and fiber volume. Both the bending and tensile strengths of type I were significantly higher than those of type II in the corresponding parts, indicating a correlation between the laminated structure and physical properties of BLCs. The influence of the fiber volume on BLC performance was complex. An increase in fiber volume enhanced the strength of the bamboo structure. Conversely, excessive fiber volume weakened the strength of the microstructural bonds between the fibers and resin. By combining high fiber volume and cross lamination, some physical properties of the BLCs may be reduced, as shown in the tensile test of type II-H. Results showed that type I-L had the highest bending strength (145.1 MPa). However, type I-M had the highest tensile strength (101.53 MPa), which indicates that the effect of increasing fiber content on mechanical properties is not always positive. Cross lamination effectively reduced the anisotropy of the BLCs and improved water resistance, but the inhibition of thickness expansion was minimal. In addition, varying the fiber volume also resulted in varying water resistance values among the tested BLC samples by affecting bonding strength. Types I-M and II-L showed the lowest *TSR* and *WSR* (0.56% and 0.07%, respectively). The expansion of width and thickness can be relatively controlled by adjusting the lamination method and fiber volume content. Therefore, we conclude that laminated structures and veneers with varying fiber volume contents should be chosen according to the specific application scenarios. Exploration of the novel multilayer BLC derived from bamboo veneer is beneficial to the graded utilization of fiber gradient bamboo due to its great potential for both structural and decorative applications.

## Figures and Tables

**Figure 1 polymers-14-04820-f001:**
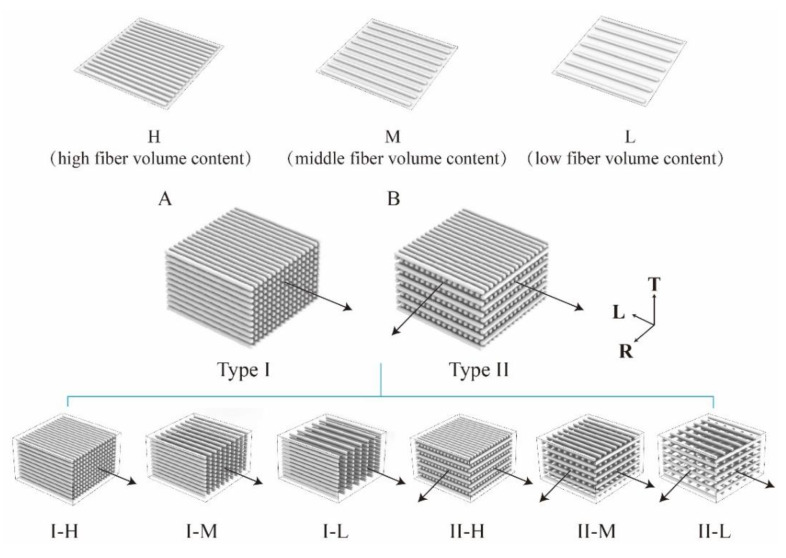
Experimental laminated structure: (**A**) type I: parallel lamination with varying fiber volume; (**B**) type II: cross lamination with varying fiber volume. Arrows indicate the direction of grain in the veneer layers; T, thickness; L, length; and R, width. I-H, parallel lamination with high fiber volume content composites; I-M, parallel lamination with middle fiber volume content composites; I-I, parallel lamination with low fiber volume content composites; II-H, cross lamination with high fiber volume content composites; II-M, cross lamination with middle fiber volume content composites; II-I, cross lamination with low fiber volume content composites.

**Figure 2 polymers-14-04820-f002:**
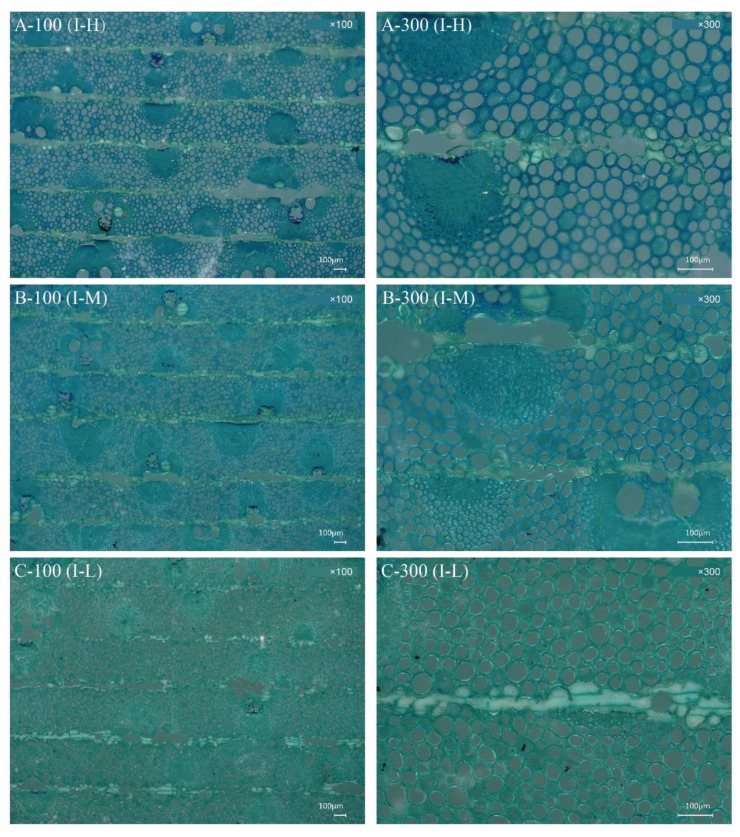
Photomicrographs of transverse sections of bamboo laminated composites of parallel lamination (type I). Resin penetration status of composites with varying fiber volumes. (**A**–**C**) denote types with high (I-H), medium (I-M), and low (I-L) fiber volumes, respectively; ×100 and ×300 represent magnifications; scale bar = 100 μm. A-100, parallel lamination with high fiber volume content composites at ×100 represent magnifications; A-300, parallel lamination with high fiber volume content composites at ×300 represent magnifications; B-100, parallel lamination with middle fiber volume content composites at ×100 represent magnifications; B-300, parallel lamination with high fiber volume content composites at ×300 represent magnifications; C-100, parallel lamination with low fiber volume content composites at ×100 represent magnifications; C-300, parallel lamination with low fiber volume content composites at ×300 represent magnifications.

**Figure 3 polymers-14-04820-f003:**
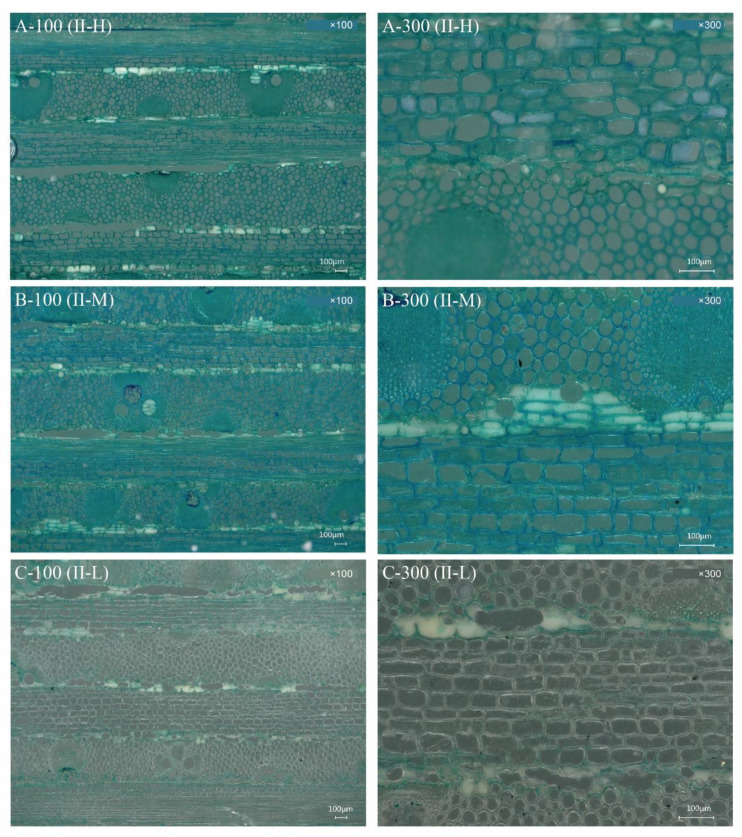
Photomicrographs of transverse sections of bamboo laminated composites of cross lamination (type II). Resin penetration status of composites with varying fiber volumes. (**A**–**C**) denote types with high (II-H), medium (II-M), and low (II-L) fiber volumes, respectively; ×100 and ×300 represent magnifications; scale bar = 100 μm. A-100, cross lamination with high fiber volume content composites at ×100 represent magnifications; A-300, cross lamination with high fiber volume content composites at ×300 represent magnifications; B-100, cross lamination with middle fiber volume content composites at ×100 represent magnifications; B-300, cross lamination with high fiber volume content composites at ×300 represent magnifications; C-100, cross lamination with low fiber volume content composites at ×100 represent magnifications; C-300, cross lamination with low fiber volume content composites at ×300 represent magnifications.

**Figure 4 polymers-14-04820-f004:**
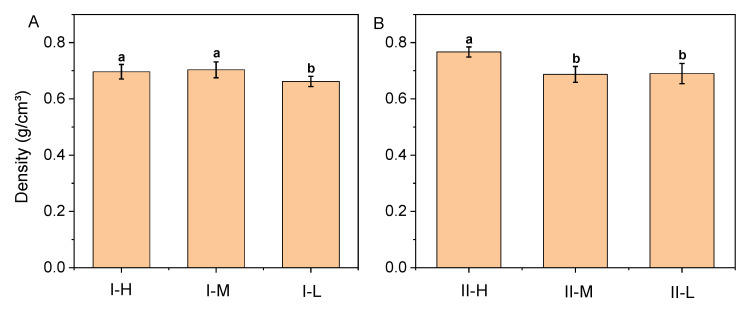
Average density of bamboo laminated composites: (**A**) parallel lamination (type I); and (**B**) cross lamination (type II). H, M, and L represent high, medium, and low fiber volumes, respectively, in the two types. Different letters within a column indicate significant differences as determined by Duncan’s multiple range test (*p* < 0.05).

**Figure 5 polymers-14-04820-f005:**
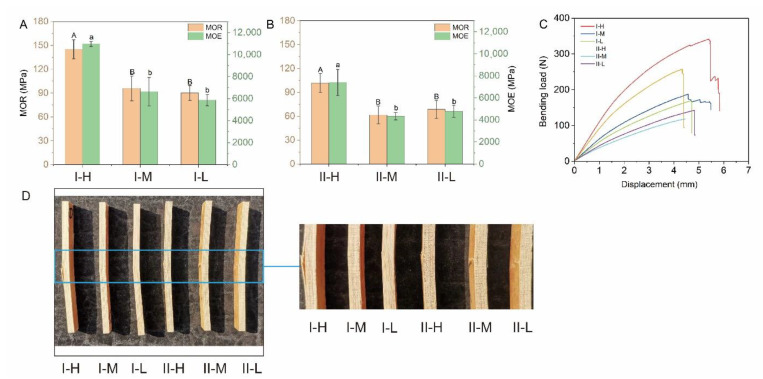
Bending responses of the various types of bamboo laminated composites (BLCs): (**A**) *MOR*; (**B**) *MOE*; (**C**) bending load–displacement curves; and (**D**) observed modes of failure and their components. Upper- and lower-case letters within the columns indicate significant differences as determined by Duncan’s multiple range test (*p* < 0.05).

**Figure 6 polymers-14-04820-f006:**
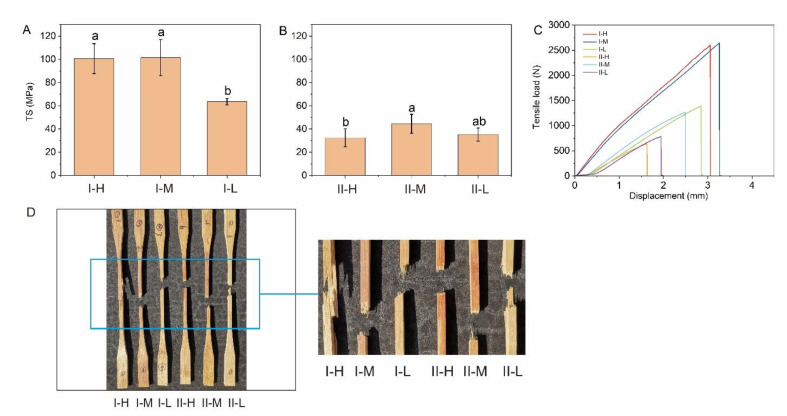
(**A**,**B**) Tensile properties of bamboo laminated composites (BLCs) by type; (**C**) tensile load–displacement curves of the distinct types of BLC; and (**D**) observed modes of failure and their components. Different letters within a column indicate significant differences as determined by Duncan’s multiple range test (*p* < 0.05).

**Figure 7 polymers-14-04820-f007:**
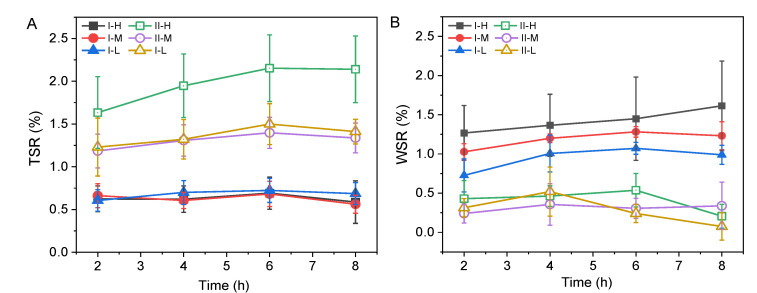
Water resistance of bamboo laminated composites (BLCs). (**A**), the *TSR* of BLCs; (**B**), the *WSR* of BLCs. Type I represents parallel laminated BLC, and type II represents cross laminated BLC. H, M, and L represent high, medium, and low fiber volumes, respectively.

## Data Availability

Not applicable.

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
