# Peer review of "Physical and Mechanical Properties of Novel Multilayer Bamboo Laminated Composites Derived from Bamboo Veneer"

_polymers, 2022, doi:10.3390/polym14224820_

Round 1

Reviewer 1 Report

The manuscript on “Physical and mechanical properties of novel multilayer bamboo laminated composites derived from bamboo veneer” brings new experimental data on an interesting topic.

A large variety of materials are prepared from bamboo and a wide range of techniques are used for their analyses. The presentation of data does not reflect the conclusions. The abstract and the conclusion need a rewriting accordingly. I also recommend to the authors to present quantitative discussions.

Note also that images on Figs  2 and 3 do nor bring new information.

Author Response

To Reviewer #1,

Dear professor,

We deeply appreciate the valuable suggestion about the manuscript (MS). Indeed, these comments are very useful for us to further improve the MS. Now we complete the revision of this work. We do hope you think this new version of the MS is satisfactory for publication.

There are 2 items in the comment in total. With high respect to you, we would be delighted to answer your point to point as follows:

Reviewers' comments:

Reviewer #1: The manuscript on “Physical and mechanical properties of novel multilayer bamboo laminated composites derived from bamboo veneer” brings new experimental data on an interesting topic.

A large variety of materials are prepared from bamboo and a wide range of techniques are used for their analyses. The presentation of data does not reflect the conclusions. The abstract and the conclusion need a rewriting accordingly. I also recommend to the authors to present quantitative discussions.

  1. Response: Thanks for the nice comments. We rewrote the abstract and conclusion according to your comment. A quantitative analysis was made in the discussion. Both the bending and tensile strengths of the cross lamination were lower than those of the parallel lamination in the corresponding parts. At parallel lamination, BLC made of veneers with high and middle fiber volume contents has the maximum bending and tensile strength, which are 145.1 MPa and 101.53 MPa respectively. Parallel and cross lamination inhibited expansion in the thickness and width directions, respectively. Fiber volume content also affected the bonding strength between layers of veneers so as to affect water resistance. The thickness swelling rate (TSR) and Width swelling rate (WSR) of the BLC after 8 h of treatment ranged from 0.56 to 2.14%, and the final WSR ranged from 0.07 to 1.61% respectively. Please check it,

Note also that images on Figs. 2 and 3 do not bring new information.

  1. Response: Thanks for the comments. We try to confirm the penetration of resin by Figs. 2 and 3. The gradient distribution of vascular bundles resulted in different probabilities of resin filling and formation of a covering layer in the cell cavity of the parenchyma cells to form a cell wall polymer. Moreover, in the cross lamination, the fibers in the profile maintained a long distance in the bonding interface, resulting in a severe lack of resin compared to the other types. These phenomena all reflect that the laminated structure and fiber volume contents significantly influenced the final properties of the BLC.

Again, the authors wish to thank the reviewers for their thoughtful comments and their work on our manuscript.

With best regards

Sincerely yours,

Fei Rao*, [email protected]

Zhang Weizhong, [email protected]

School of Art and Design, Zhejiang Sci-Tech University, No. 2 street 928, Hangzhou 310018, China

Reviewer 2 Report

We put the review in the attachment.

Author Response

To Reviewer #2,

Dear professor,

Here we must express our deep gratitude firstly to you for spending time to evaluate the publication of this manuscript (MS) and the visionary comments, which are of great help to guide our follow-up research work.

Reviewer #2: We received paper from Polymers MDPI with the title “Physical and mechanical properties of novel multilayer bamboo laminated composites derived from bamboo veneer”

Before it goes to accepted, several points need to be clarified and revised.

  1. This type of paper is research paper or review? If this is review, all figures that used from previous study should be cited. The references also need to enlarge since 36 references is too low for review paper.

Response: Thanks for the comments. We feel really sorry for our carelessness. The type of paper is research paper. We corrected the type in manuscript according to your comment. Please check it.

  1. Abstract should be revised. If its research paper, put the most important results in the value (number) mode in abstract.

Response: Thanks for the comments. We rewrote the abstract according to your comment. Both the bending and tensile strengths of the cross lamination were lower than those of the parallel lamination in the corresponding parts. At parallel lamination, BLC made of veneers with high and middle fiber volume contents has the maximum bending and tensile strength, which are 145.1 MPa and 101.53 MPa respectively. Parallel and cross lamination inhibited expansion in the thickness and width directions, respectively. Fiber volume content also affected the bonding strength between layers of veneers so as to affect water resistance. The thickness swelling rate (TSR) and Width swelling rate (WSR) of the BLC after 8 h of treatment ranged from 0.56 to 2.14%, and the final WSR ranged from 0.07 to 1.61% respectively. Please check it.

  1. In the introduction, composites need to be listed from GFRP, CFRP, Natural fibres. Before goes to BLC. The present research can be used to strengthen the introduction.

https://doi.org/10.3390/polym14204322, https://doi.org/10.1016/j.jmrt.2021.07.152, https://doi.org/10.1016/j.compstruct.2021.113707

Response: Thanks for the nice comments. Your suggestion is of great help to our research progress. We have added the classification of composite materials to the introduction in line 51 according to your advice. Please check it.

  1. In Fig. 5 as shown below

Why A and B is not put in 1 graph? Since these bending results, and have same standard test?

Response: Thanks for the comments. We are very sorry for the trouble caused to your review due to our lack of experience. Only one variable can be controlled when using SPSS for analysis. We try to make it easier for readers to understand the significant differences in experimental results by categorizing them. A and B represent the bending strength of different fiber contents under parallel and cross stratification, respectively. By comparing the properties of samples with corresponding fiber volume content of A and B, the influence of laminated structure on the bending strength of BLC can be obtained.

  1. In Fig. 6 as shown below

Why A and B is not put in 1 graph? Since these tensile results, and have same standard test?

Response: Thanks for the comments. We apologize for the inconvenience caused to your review. Only one variable can be controlled when using SPSS for analysis. A and B represent the tensile strength of different fiber contents under parallel and cross stratification, respectively. By comparing the properties of samples with corresponding fiber volume content of A and B, the influence of laminated structure on the tensile strength of BLC can be obtained.

  1. In Fig. 6 as shown below

Please reduce the number of graph. Put representative grap of each category is enough.

Response: Thanks for the comments. We choose representative samples in Figs.5 and 6 according to your advice. Please check it.

  1. In Fig. 7 as shown below

Why after 4 hours, II-H and II-L, the trend decrease? Explain more on it.

Response: Thanks for the comments. This might be because in the cross lamination of type II, the adjacent veneer force direction is vertical and cancels one another, reducing contraction expansion and deformation in the width direction. The composites swelling due to water absorption is reduced.

  1. In conclusion, put all the results that important and make it compact. The author can make it bullet based or in 1 paragraph.

Response: Thanks for the comments. We highlighted the important data in conclusion according to your advice. The final results showed that the I-O had the highest bending strength, which are 145.1 MPa. However, I-M had the highest tensile strength (101.53 MPa), which indicate that the effect of increasing fiber content on mechanical properties is not always positive. Regarding water resistance, the I-M and II-L showed the lowest TSR and WSR, which are 0.56% and 0.07% respectively. The expansion of width and thickness can be relatively controlled by adjusting the lamination method and fiber volume content reasonably. Please check it.

Again, the authors wish to thank the reviewers for their thoughtful comments and their work on our manuscript.

With best regards

Sincerely yours,

Fei Rao*, [email protected]

Zhang Weizhong, [email protected]

School of Art and Design, Zhejiang Sci-Tech University, No. 2 street 928, Hangzhou 310018, China

Reviewer 3 Report

Overall, this paper present a good finding that is worth for publication, however, some improvement is necessary before this paper is ready to be published.

Abstract:

-        General introduction of the study should be provided prior to the objective of study.

-        The information provided in the abstract should be more specific, author might want to highlight some important data in the abstract such as the optimum strength of the composites.

Introduction:

-        At the end of introduction, author should highlight the gap of study and the interest which trigger this research to be carried out.

-         

Materials and methodology:

-        Please describe the natural drying method in more specific way.

-        Please be more specific on the difference between low, middle, and high fiber volume, such as the percentage of fiber, or the spacing difference, otherwise, this experiment is hard to be reproduced in future.

-        Please define all the alphabet in the equation, some equation is redundantly using same alphabet. For example, letter b was used in equation 2,3,4, and 5.

-        Please state the name and country for all the machine use in this study.

-        Grammar and punctuation should be revise accordingly.

Results and Discussion

-        Figure 5 (d) should be enlarged since it is an interesting finding but were compressed into a small picture.

-        Author should ensure that every argument made were supported by appropriate reference, and similar findings from previous study (if any).

-         

-         

Conclusion

-        Please highlight the significant results of this study. Such as the highest strength achieved by the composites.

Author Response

To Reviewer #3,

Dear professor,

Here we must express our deep gratitude firstly to you for spending time to evaluate the publication of this manuscript (MS) and the visionary comments, which are of great help to guide our follow-up research work.

Reviewer #3: Overall, this paper present a good finding that is worth for publication, however, some improvement is necessary before this paper is ready to be published.

Abstract:

-        General introduction of the study should be provided prior to the objective of study.

Response: Thanks for the comments. We provided general introduction prior to the objective of study in abstract. Please check it.

-        The information provided in the abstract should be more specific, author might want to highlight some important data in the abstract such as the optimum strength of the composites.

Response: Thanks for the comments. We rewrote the abstract and highlight important data according to your comment. Both the bending and tensile strengths of the cross lamination were lower than those of the parallel lamination in the corresponding parts. At parallel lamination, BLC made of veneers with high and middle fiber volume contents has the maximum bending and tensile strength, which are 145.1 MPa and 101.53 MPa respectively. Parallel and cross lamination inhibited expansion in the thickness and width directions, respectively. Fiber volume content also affected the bonding strength between layers of veneers so as to affect water resistance. The thickness swelling rate (TSR) and Width swelling rate (WSR) of the BLC after 8 h of treatment ranged from 0.56 to 2.14%, and the final WSR ranged from 0.07 to 1.61% respectively. Please check it.

Introduction:

-        At the end of introduction, author should highlight the gap of study and the interest which trigger this research to be carried out.

Response: Thanks for the comments. We highlighted the gap of study and the interest which trigger this research to be carried out at the end of introduction. Previous research has mainly focused on the preparation of laminated veneer lumber (LVL) and plywood derived from parallel and cross lamination wood veneers, respectively. Several studies have shown that the physical and mechanical properties of LVL and plywood are influenced by the quality of the veneer, wood species, number and order of layers, and the adhesive used for bonding. At present, as far as we know there are relative few research on the influence of fiber volume content on the properties of laminated composites. Rational utilization of the special fiber gradient structure of bamboo will achieve better resource allocation. Please check it.

Materials and methodology:

-        Please describe the natural drying method in more specific way.

Response: Thanks for the comments. The bamboo veneers were air-dried to a moisture content (MC) about 12% under the ambient environment.

-        Please be more specific on the difference between low, middle, and high fiber volume, such as the percentage of fiber, or the spacing difference, otherwise, this experiment is hard to be reproduced in future.

Response: Thanks for the excellent comments. On the treatment of high, meddle and low fiber volume content method, we referred to the relevant literature [1]. In the process of rotary cutting, bamboo culm was divided evenly into 3 slices from the inner (IB), middle (MB), and outer (OB) bamboo layers along the thickness of the culm.

-        Please define all the alphabet in the equation, some equation is redundantly using same alphabet. For example, letter b was used in equation 2,3,4, and 5.

Response: Thanks for the comments. We redefined all the alphabet in the equation. Please check it.

-        Please state the name and country for all the machine use in this study.

Response: Thanks for the comments. We are very sorry for the trouble caused to your review due to our lack of experience. We added the name and country for all the machine use in this study.

-        Grammar and punctuation should be revise accordingly.

Response: Thanks for the comments. We revised the grammar and punctuation. And submitted to a professional academic editor to re-proofread the full text to reduce some ambiguities, errors and expressions. Please check it.

Results and Discussion

-        Figure 5 (d) should be enlarged since it is an interesting finding but were compressed into a small picture.

Response: Thanks for the comments. We enlarged the Figure 5 (d) according to your advice. Please check it.

-        Author should ensure that every argument made were supported by appropriate reference, and similar findings from previous study (if any).

Response: Thanks for the comments. We added appropriate reference to support our argument in results and discussion. Please check it.

Conclusion

-        Please highlight the significant results of this study. Such as the highest strength achieved by the composites.

Response: Thanks for the comments. The final results showed that the I-O had the highest bending strength, which are 145.1 MPa. However, I-M had the highest tensile strength (101.53 MPa), which indicate that the effect of increasing fiber content on mechanical properties is not always positive. Regarding water resistance, the I-M and II-L showed the lowest TSR and WSR, which are 0.56% and 0.07% respectively. The expansion of width and thickness can be relatively controlled by adjusting the lamination method and fiber volume content reasonably. Please check it.

Reference

  1. Wei, X.; Wang, G.; Smith, L.M.; Jiang, H. The Hygroscopicity of Moso Bamboo (Phyllostachys Edulis) with a Gradient Fiber Structure. J. Mater. Res. Technol. 2021, 15, 4309–4316, doi:10.1016/j.jmrt.2021.10.038.

Again, the authors wish to thank the reviewers for their thoughtful comments and their work on our manuscript.

With best regards

Sincerely yours,

Fei Rao*, [email protected]

Zhang Weizhong, [email protected]

School of Art and Design, Zhejiang Sci-Tech University, No. 2 street 928, Hangzhou 310018, China

Reviewer 4 Report

According to the manuscript with title: "Physical and mechanical properties of novel multilayer bam- 2 boo laminated composites derived from bamboo veneer". The submitted work is introducing a new valuable and interesting idea and the given results confirm the idea. This work is suitable for publication in the Journal. I suggest the acceptance after some major corrections as follows;

1.     Abstract section need to rewrite in correct sequence with more explanation

2.     What is the application of this study ?

3.     Reformulate the aim of the work in introduction

4.     Add previous published work with comparison to clear the novelty of your work

5.     Add critical results with numbers in abstract section

6.     Give some results with numbers in conclusion

7.     Some words are in cross-linked with others

8.     The caption of all figures is very thin, need to add more details in each caption

9.     Add more explanation to experimental work

10.  Correct typographical errors.

11.  Don't use abbreviations in title and abstract, you must define it in first time use

Author Response

To Reviewer #4,

Dear professor,

We deeply appreciate the valuable suggestion about the manuscript (MS). Indeed, these comments are very useful for us to further improve the MS. Now we complete the revision of this work. We do hope you think this new version of the MS is satisfactory for publication.

There are 11 items in the comment in total. With high respect to you, we would be delighted to answer your point to point as follows:

Reviewers' comments:

Reviewer #4: According to the manuscript with title: "Physical and mechanical properties of novel multilayer bamboo laminated composites derived from bamboo veneer". The submitted work is introducing a new valuable and interesting idea and the given results confirm the idea. This work is suitable for publication in the Journal. I suggest the acceptance after some major corrections as follows;

  1. Abstract section need to rewrite in correct sequence with more explanation

Response: Thanks for the comments. We rewrote the abstract according to your comment. We provided general introduction prior to the objective of study and added critical results with numbers in abstract section in abstract. Please check it.

  1. What is the application of this study?

Response: Thanks for the nice comments. This study provides a reference for the selection of an appropriate laminated structure and fiber volume content of bamboo laminated composites made from bamboo veneers in future practical applications of decoration and structure. For example, high strength composites made from high fiber volume content are more suitable for construction, low strength composites made from low fiber volume content are more suitable for decoration. So as to realize the reasonable allocation of resources.

  1. Reformulate the aim of the work in introduction

Response: Thanks for the comments. We rewrote the aim of the work in introduction. Please check it.

  1. Add previous published work with comparison to clear the novelty of your work

Response: Thanks for the nice comments. We added some previous published work with comparison to clear the novelty of our work in the end of introduction. Previous research has mainly focused on the preparation of laminated veneer lumber (LVL) and plywood derived from parallel and cross lamination wood veneers, respectively. Several studies have shown that the physical and mechanical properties of LVL and plywood are influenced by the quality of the veneer, wood species, number and order of layers, and the adhesive used for bonding. At present, as far as we know there are relative few research on the influence of fiber volume content on the properties of laminated composites. Rational utilization of the special fiber gradient structure of bamboo will achieve better resource allocation. Please check it.

  1. Add critical results with numbers in abstract section

Response: Thanks for the comments. We rewrote the abstract and highlight important data according to your comment. Both the bending and tensile strengths of the cross lamination were lower than those of the parallel lamination in the corresponding parts. At parallel lamination, BLC made of veneers with high and middle fiber volume contents has the maximum bending and tensile strength, which are 145.1 MPa and 101.53 MPa respectively. Parallel and cross lamination inhibited expansion in the thickness and width directions, respectively. Fiber volume content also affected the bonding strength between layers of veneers so as to affect water resistance. The thickness swelling rate (TSR) and Width swelling rate (WSR) of the BLC after 8 h of treatment ranged from 0.56 to 2.14%, and the final WSR ranged from 0.07 to 1.61% respectively. Please check it.

  1. Give some results with numbers in conclusion

Response: Thanks for the comments. We highlighted important data in conclusion according to your comment. The final results showed that the I-O had the highest bending strength, which are 145.1 MPa. However, I-M had the highest tensile strength (101.53 MPa), which indicate that the effect of increasing fiber content on mechanical properties is not always positive. Regarding water resistance, the I-M and II-L showed the lowest TSR and WSR, which are 0.56% and 0.07% respectively. The expansion of width and thickness can be relatively controlled by adjusting the lamination method and fiber volume content reasonably.Please check it.

  1. Some words are in cross-linked with others

Response: Thanks for the comments. We are very sorry for the trouble caused to your review due to our lack of experience. We submitted to a professional academic editor to re-proofread the full text to reduce some ambiguities, errors and expressions. Please check it.

  1. The caption of all figures is very thin, need to add more details in each caption

Response: Thanks for the comments. We added more details in each caption in Figs. 1, 2, 3, 5 and 6. Please check it.

  1. Add more explanation to experimental work

Response: Thanks for the comments. We explained more details about result of water resistance according to your comment. After 4 hours, the WSR of II-H and II-L showed a downward trend. This might be cross lamination can reduce the deformation caused by anisotropy in width direction. The difference of WSR was reduced by cross lamination which inhibited the contraction and deformation of width direction. In addition, by increasing temperature and time exposure, the phenomenon of absorption and expansion of bamboo can be reduced.

  1. Correct typographical errors.

Response: Thanks for the comments. We corrected typographical errors according to your advice. The type of paper is research paper. We corrected the type in manuscript. Please check it.

  1. Don't use abbreviations in title and abstract, you must define it in first time use

Response: Thanks for the comments. We got rid of the abbreviations in title and abstract according to your advice. Please check it.

Again, the authors wish to thank the reviewers for their thoughtful comments and their work on our manuscript.

With best regards

Sincerely yours,

Fei Rao*, [email protected]

Zhang Weizhong, [email protected]

School of Art and Design, Zhejiang Sci-Tech University, No. 2 street 928, Hangzhou 310018, China

Round 2

Reviewer 4 Report

Accept the paper in this form